# Genome-wide association study in 176,678 Europeans reveals genetic loci for tanning response to sun exposure

Alessia Visconti [1], David L. Duffy[2], Fan Liu [3,4,5], Gu Zhu[2], Wenting Wu[6], Yan Chen[3,4], Pirro G. Hysi [1], Changqing Zeng[3], Marianna Sanna[1], Mark M. Iles[7], Peter A. Kanetsky[8], Florence Demenais[9,10], Merel A. Hamer[11], Andre G. Uitterlinden[12,13], M. Arfan Ikram [13], Tamar Nijsten[11], Nicholas G. Martin[2], Manfred Kayser [5], Tim D. Spector[1], Jiali Han[6,14], Veronique Bataille[1,15] & Mario Falchi[1]

The skin's tendency to sunburn rather than tan is a major risk factor for skin cancer. Here we report a large genome-wide association study of ease of skin tanning in 176,678 subjects of European ancestry. We identify significant association with tanning ability at 20 loci. We confirm previously identified associations at six of these loci, and report 14 novel loci, of which ten have never been associated with pigmentation-related phenotypes. Our results also suggest that variants at the *AHR/AGR3* locus, previously associated with cutaneous malignant melanoma the underlying mechanism of which is poorly understood, might act on disease risk through modulation of tanning ability.

[1] Department of Twin Research and Genetic Epidemiology, King's College London, London SE1 7EH, UK. [2] QIMR Berghofer Medical Research Institute, Brisbane 4029, Australia. [3] CAS Key Laboratory of Genomic and Precision Medicine, Beijing Institute of Genomics, Chinese Academy of Sciences, Beijing 100101, China. [4] University of Chinese Academy of Sciences, Beijing 100049, China. [5] Department of Genetic Identification, Erasmus MC University Medical Center Rotterdam, Rotterdam 3000 CA, The Netherlands. [6] Department of Epidemiology, Richard M. Fairbanks School of Public Health, Melvin & Bren Simon Cancer Center, Indiana University, Indianapolis 46202 IN, USA. [7] Section of Epidemiology and Biostatistics, Leeds Institute of Cancer and Pathology, University of Leeds, Leeds LS9 7TF, UK. [8] Department of Cancer Epidemiology, H. Lee Moffitt Cancer Center and Research Institute, Tampa 33612 FL, USA. [9] INSERM, UMR 946, Genetic Variation and Human Diseases Unit, Paris 75010, France. [10] Institut Universitaire d'Hématologie, Université Paris Diderot, Sorbonne Paris Cité, Paris 75010, France. [11] Department of Dermatology, Erasmus MC University Medical Center Rotterdam, Rotterdam 3000 CA, The Netherlands. [12] Department of Internal Medicine, Erasmus MC University Medical Center Rotterdam, Rotterdam 3000 CA, The Netherlands. [13] Department of Epidemiology, Erasmus MC University Medical Center Rotterdam, Rotterdam 3000 CA, The Netherlands. [14] Channing Division of Network Medicine, Department of Medicine, Brigham and Women's Hospital, Boston 02115 MA, USA. [15] Department of Dermatology, West Herts NHS Trust, Herts HP2 4AD, UK. Correspondence and requests for materials should be addressed to M.F. (email: mario.falchi@kcl.ac.uk)

Sun exposure has consistently been associated with increased risk of all skin cancers, including cutaneous malignant melanoma (CMM), basal cell carcinoma and squamous cell carcinoma[1]. These are the most common types of cancer in European populations of which the incidence rate is higher in fair-skinned people, rather than darker-skinned people. The tanning response after exposure to sunlight is mainly determined by melanin pigmentation, and aims at protecting the skin from DNA photodamage. Tanning response shows large variability within and between populations. The heritability of ease of skin tanning attributable to common genetic variation in the UK Biobank sample [2] has been estimated to be 0.454 ± 0.006.

Genome-wide association studies (GWASs) on European populations have identified several DNA variants involved in tanning ability or in skin sensitivity to sunlight, encompassing seven genes, namely *ASIP*, *EXOC2*, *HERC2*, *IRF4*, *MC1R*, *SLC45A2* and *TYR*[3–6], that are already known to be associated with both pigmentation-related traits, such as hair, eye or skin colour[6–10], and skin cancer[11–13]. However, it has already been observed than while some loci exert an effect on both pigmentation and tanning ability, others have a more specific effect[14]. This also suggests two pathways for skin cancer development: via pigmentation or independent of pigmentation[14].

To further investigate the genetic basis of skin tanning in Europeans and their effect on skin cancer susceptibility, we perform here a large-scale GWAS using data from the UK Biobank[15], identifying ten novel associations, and replicating ten genes previously associated with ease of skin tanning or pigmentation-related phenotypes. Additionally, we show a genetic correlation between ease of skin tanning and non-melanoma skin cancer, and

highlight shared genetic effects between variants at *AHR/AGR3* and tanning ability and CMM risk.

## Results

**Genome-wide association analyses**. Ease of skin tanning and genotype data were available for 121,296 individuals of European ancestry from the UK Biobank[15] (UKBB), which were divided in two groups according to their skin's ability to tan, with 38.6% of the individuals reporting that they never tan and only burn, or get mildly or occasionally tanned (Methods; Supplementary Table 1).

We carried out a genome-wide association analysis at 8,351,141 SNPs, assuming an additive genetic model, with sex included as covariate, and applying PCA-based correction to address potential population stratification (Methods; Supplementary Fig. 1). We identified 10,834 SNPs passing genome-wide significance ($P < 5.0 \times 10^{-8}$) that mapped to 30 distinct loci (Fig. 1, Supplementary Table 2, Supplementary Data 1 and Supplementary Fig. 2 and 3). The genetic inflation factor $\lambda_{GC}$ was 1.10. Therefore, to rule out the possibility that some associations could be driven by confounding biases, we used the LD score regression approach[16], which confirmed the presence of an underlying polygenic architecture (intercept = 1.04 ± 0.01).

We attempted to replicate these 30 loci in five additional cohorts of European ancestry for which data on tanning ability were available ($N = 55,382$; Methods; Supplementary Tables 3–8). Meta-analysis of the results in these five replication cohorts confirmed, at a Bonferroni-corrected threshold of $0.05/30 = 1.67 \times 10^{-3}$, the association at 20 of the 30 top-associated SNPs at each locus. The replicated loci encompass six genes previously

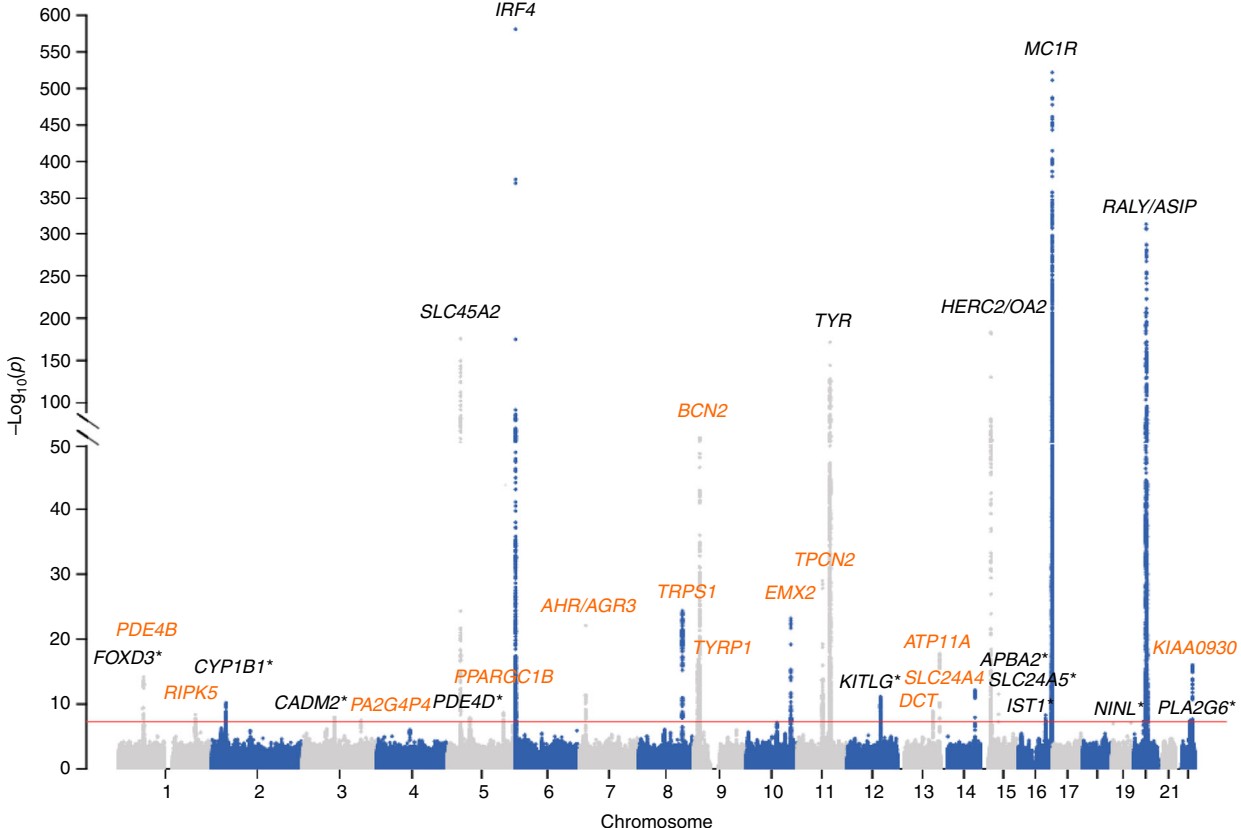

**Fig. 1** Manhattan plot of ease of skin tanning results in the UKBB data set. The *P* values were obtained by logistic regression analysis assuming an additive genetic model with sex and the first five principal components of the genotype data as covariates. The *x*-axis shows the genomic coordinates (GRCh37.p13) of the tested SNPs and the *y*-axis shows the −log₁₀ *P* value of their association. The horizontal red line indicates the threshold for genome-wide significance at $5.0 \times 10^{-8}$. The ten genes that failed replication are indicated by *; genes newly associated with tanning ability are reported in orange

**Table 1 Genome-wide association and replication for ease of skin tanning**

| SNP | CHR:BP | EA | MAF | OR (95% CI) | SE | $P_{UKBB}$ | $P_{replication}$ | Gene | Note |
|---|---|---|---|---|---|---|---|---|---|
| rs1308048 | 1:66888542 | C | 0.42 | 0.93 (0.92–0.95) | 0.01 | $2.09 \times 10^{-14}$ | $2.83 \times 10^{-8}$ | PDE4B | a |
| rs12078075 | 1:205163798 | G | 0.09 | 1.09 (1.06–1.13) | 0.02 | $3.99 \times 10^{-9}$ | $1.71 \times 10^{-4}$ | RIPK5 | a |
| rs9818780 | 3:156492758 | C | 0.49 | 1.05 (1.03–1.07) | 0.01 | $3.42 \times 10^{-8}$ | $1.10 \times 10^{-5}$ | PA2G4P4 | a |
| rs16891982 | 5:33951693 | C | 0.03 | 0.40 (0.03–0.38) | 0.03 | $2.02 \times 10^{-176}$ | $2.45 \times 10^{-36}$ | SLC45A2 | b |
| rs251464 | 5:149196234 | C | 0.25 | 0.94 (0.92–0.96) | 0.01 | $2.16 \times 10^{-9}$ | $2.79 \times 10^{-8}$ | PPARGC1B | a |
| rs12203592 | 6:396321 | T | 0.22 | 1.74 (1.69–1.76) | 0.01 | $1.05 \times 10^{-581}$ | $4.40 \times 10^{-157}$ | IRF4 | b |
| rs117132860 | 7:17134708 | A | 0.03 | 1.30 (1.23–1.36) | 0.03 | $7.63 \times 10^{-23}$ | $2.93 \times 10^{-4}$ | AHR/AGR3 | a |
| rs2737212 | 8:116621214 | C | 0.45 | 1.09 (1.08–1.11) | 0.01 | $4.33 \times 10^{-25}$ | $4.80 \times 10^{-9}$ | TRPS1 | a |
| rs1326797 | 9:12716762 | T | 0.37 | 0.93 (0.91–0.94) | 0.01 | $1.24 \times 10^{-17}$ | $2.62 \times 10^{-11}$ | TYRP1 | c |
| rs10810650 | 9:16873551 | C | 0.39 | 0.87 (0.85–0.88) | 0.01 | $2.38 \times 10^{-59}$ | $2.10 \times 10^{-29}$ | BNC2 | c |
| rs35563099 | 10:119572403 | T | 0.16 | 0.89 (0.87–0.91) | 0.01 | $6.61 \times 10^{-24}$ | $5.33 \times 10^{-7}$ | EMX2 | a |
| rs72917317 | 11:68817441 | G | 0.10 | 1.18 (1.14–1.21) | 0.01 | $1.02 \times 10^{-29}$ | $1.31 \times 10^{-8}$ | TPCN2 | c |
| rs1126809 | 11:89017961 | A | 0.31 | 1.29 (1.21–1.32) | 0.01 | $2.42 \times 10^{-172}$ | $2.59 \times 10^{-75}$ | TYR | b |
| rs9561570 | 13:95156198 | T | 0.31 | 1.06 (1.04–1.08) | 0.01 | $1.41 \times 10^{-9}$ | $2.95 \times 10^{-7}$ | DCT | a |
| rs1046793 | 13:113539894 | C | 0.46 | 0.93 (0.91–0.94) | 0.01 | $2.00 \times 10^{-18}$ | $1.97 \times 10^{-5}$ | ATP11A | a |
| rs746586 | 14:92775967 | T | 0.45 | 1.06 (1.05–1.08) | 0.01 | $6.95 \times 10^{-13}$ | $1.17 \times 10^{-5}$ | SLC24A4 | c |
| rs12913832 | 15:28365618 | A | 0.22 | 0.74 (0.72–0.75) | 0.01 | $6.32 \times 10^{-184}$ | $2.99 \times 10^{-48}$ | HERC2/OCA2 | b |
| rs369230 | 16:89645437 | G | 0.30 | 1.60 (1.57–1.63) | 0.01 | $1.00 \times 10^{-522}$ | $8.28 \times 10^{-132}$ | MC1R | b |
| rs6059655 | 20:32665748 | A | 0.10 | 1.69 (1.65–1.74) | 0.01 | $1.44 \times 10^{-315}$ | $2.98 \times 10^{-99}$ | RALY/ASIP | b |
| rs11703668 | 22:45630335 | G | 0.46 | 0.93 (0.92–0.95) | 0.01 | $1.00 \times 10^{-16}$ | $4.42 \times 10^{-4}$ | KIAA0930 | a |

The top-associated SNP is reported at each replicated locus, along with the genomic coordinates (CHR:BP; GRCh37.p13), the effect allele (EA), the minor allele frequency (MAF), the odds ratio (OR) with its 95% confidence interval (CI) and standard error (SE), the association P value in the discovery set ($P_{UKBB}$) and the meta-analysis P value in the five independent replication cohorts ($P_{replication}$). Positive odds ratios indicate a decreased tanning ability
[a]Indicates a novel association
[b]Indicates a known association with tanning ability
[c]Indicates a known association with other pigmentation-related traits

associated with tanning ability, four genes previously associated with pigmentation-related traits, and ten novel associations (Table 1, Supplementary Tables 2 and 9, and Supplementary Fig. 4–23). Conditional association analysis highlighted 14 further independent genome-wide significant associations, which were replicated in the additional cohorts (Supplementary Table 10).

**Sex differences.** It has been repeatedly observed that adult females are fairer-skinned than males[17]. In this study, given the large sample size and the highly significant effects identified, we further investigated the sex-by-SNP interaction for the 20 loci associated with tanning ability. We identified significant interactions ($P < 2.5 \times 10^{-3}$) at five different loci in the UKBB data set, which, however, were not confirmed in the additional cohorts (Supplementary Table 11).

**Genome-wide significant loci.** Six of the identified loci included genes associated with tanning ability in previous GWASs (NHGRI-EBI GWAS Catalog, release 26 June 2017): HERC2/OCA2, IRF4, MC1R, RALY/ASIP, SLC45A2 and TYR[3–6]. Additionally, we identified four other genes whose evidence of association with pigmentation-related traits had previously been limited only to skin pigmentation (BNC2[9]), hair colour (TPCN2[4]) or both hair and eye colour (SLC24A4[3] and TYRP1[4,18]). The high statistical power provided by the large UKBB sample size (>80% for a common variant with minor allele frequency of 0.5 and odds ratio >1.06 at α-level $5 \times 10^{-8}$) allowed for the identification of low-penetrance common DNA variants of small effect size at ten novel genes. Five of these novel genes are known for their involvement in melanin synthesis: DCT (TYRP2) which catalyses melanin production;[19] EMX2 and PPARGC1B, regulators of the melanocyte-specific transcription factor (MITF)[19,20], and PDE4B and RIPK5, which have been suggested to be MITF targets[21,22]. Five associations do not harbour any obvious candidate for ease of skin tanning nor pigmentation: AHR/AGR3 ATP11A, TRPS1,

KIAA0930 and an intergenic region flanking the PA2G4P4 pseudogene, whose functional role in tanning ability would require further investigation. Interestingly, KIAA0930 encodes the Q6ICG6 protein that interacts with the tyrosine 3-monooxygenase/tryptophan 5-monooxygenase activation protein, which have been suggested to be important in melanocyte development in a mouse model[23].

Among the genes identified by the conditional analysis, namely CHMP1A, FANCA, SPIRE2, DEF8, AFG3L1P, DBNDD1 and PRDM7, DBNDD1 has been previously associated with tanning ability[5], while mutations in the FANCA gene are responsible for Fanconi anaemia, a disease characterised by short stature, bone marrow failure and increased risk of multiple cancers. Affected individuals present areas of skin hypopigmentation and café au lait spots with abnormal melanin deposition.

**Effect of ease of skin tanning variants on skin cancer.** Several of the identified genes have previously been associated with increased susceptibility to skin cancer. Common DNA variants at the AFG3L1P, AGR3, HERC2/OCA2, MC1R, RALY/ASIP, SLC45A2 and TYR genes have been linked to CMM[11,24], and common DNA variants at the AHR, ASIP, BNC2, DEF8, IRF4, MC1R, OCA2, TYR and SLC45A2 genes have been associated with non-melanoma skin cancer risk[12,13,25]. ATP11A was significantly associated in the first stage of a large GWAS including 12,945 self-reported basal cell carcinoma cases and 274,252 controls of European ancestry from the 23andMe database (but failed replication in the final meta-analysis)[13], with top association at rs1765871 ($P = 4.9 \times 10^{-9}$) in strong linkage disequilibrium with our top-associated SNP (rs1046793; $r^2 = 0.90$). Suggestive association with skin cancer has also been reported for TRPS1[12]. Additionally, common variants at the FANCA gene have been suggested to predict melanoma survival[26].

While the number of melanoma cases in the UKBB data set was not sufficient to generate a reliable estimate of the genetic

correlation between ease of skin tanning and CMM, we could estimate the genetic correlation of ease of skin tanning with non-melanoma skin cancer, which amounted to $\rho = 0.33$ (SD = 0.16, $P = 0.04$; Methods, Supplementary Tables 12 and 13, Supplementary Figs. 24 and 25, and Supplementary Data 2).

The present study also shows association with ease of skin tanning for the locus encompassing AGR3/AHR, which has previously been associated with CMM risk[11] but not with tanning ability nor pigmentation. Our association at AGR3/AHR overlaps with previous findings for CMM risk, with rs1721028 (secondary association at AGR3/AHR) showing $r^2 > 0.8$ with rs1636744, identified by Law et al.[11]. We further investigate whether variants at this locus exert a shared effect on tanning ability and CMM using a Bayesian bivariate approach[27] (Methods). The AGR3/AHR posterior probability for shared genetic effects between tanning ability and CMM was 1, thus indicating that the same genetic variants are involved in both decreased tanning ability and increased risk of melanoma.

**Effects of ease of skin tanning variants on hair colour.** We further investigated the effects of loci associated with tanning ability in our study on hair colour, which was also characterised in the UKBB data set. Specifically, we separately assessed the association with non-red hair colour, and with red versus non-red hair colour (Methods; Supplementary Tables 14 and 15). Most of the genes that have been previously associated with non-red hair colour[4,6,8,18] (i.e., IRF4, HERC2/OCA2, SLC24A4, SLC45A2, TPCN2 and TYRP1) showed a stronger association with non-red hair colour than with ease of skin tanning, with the exception of MC1R, RALY/ASIP and TYR (Supplementary Table 16). We additionally observed three genome-wide significant associations between non-red hair colour and BNC2 (previously associated with freckles[8] and facial pigmentation[9]), DCT, and RIPK5, which were confirmed and replicated in a large meta-analysis study[28]. A marginal association was observed at KIAA0930. Red hair showed significant association with IRF4, HERC2/OCA2, MC1R and RALY/ASIP, while a marginal association was observed at SLC45A2 (Supplementary Table 17). Only MC1R was more strongly associated with red hair colour than with ease of skin tanning. Overall, seven loci were exclusively associated with tanning ability (AHR/AGR3, ATP11A, EMX2, PA2G4P4, PDE4B, PPARGC1B and TRPS1).

**Effect of MC1R variants on tanning ability and hair colour.** To better characterise the MC1R gene, which plays central roles in pigmentation and in the melanocyte response to UV exposure, we took advantage of the large UKBB sample size and tested the association between the nine most studied MC1R variants[29] present in our panel (D84E, D294H, I155T, R142H, R151C, R160W, R163Q, V60L and V92M) and both ease of skin tanning and natural hair colour. Among these, the NHGRI-EBI GWAS catalogue reports association between R151C and ease of skin tanning, freckles, hair and skin colour[3,6], and both melanoma[30] and non-melanoma skin cancers[12,13]. A haplotype, including multiple MC1R variants, was also associated with skin pigmentation[10]. In the UKBB data set, all variants apart from I155T, were strongly associated with red hair colour, and all variants but R163Q were associated with non-red hair (Supplementary Table 18). Moreover, all the studied nine variants but two (R163Q and V92M) were highly significantly associated with ease of skin tanning (Supplementary Table 19). Interestingly, these variants at the MC1R gene completely explained our top-associated signal at rs369230, which changed from $P = 1.00 \times 10^{-522}$ to 0.37.

**Functional enrichment analyses.** We defined an expanded set of SNPs including the replicated SNPs and those in strong linkage disequilibrium with them ($r^2 \geq 0.8$, N = 599). This set was enriched (empirical $P < 0.05$) for cis-eQTLs identified at 5% FDR in sun-exposed skin tissue and fibroblast, but not in non-sun exposed skin tissue from the GTEx consortium project[31] (Methods; Supplementary Table 20). Interestingly, DNA variants at the locus harbouring the MC1R gene regulate, among others, in sun-exposed skin tissue, the SPATA33 gene, that has been associated with facial pigmentation[9] and cutaneous squamous cell carcinoma[25], and, in all tissues, the CDK10 gene, that has been associated with CMM[24] (Supplementary Data 3).

The expanded set of replicated SNPs was also enriched (empirical $P < 0.05$) for regulatory elements from the RoadMap consortium project[32] (Methods; Supplementary Table 21). Specifically, we found a significant enrichment for the enhancer marks H3K36me3 and H3K27ac, which are associated with actively transcribed regions; H3K4me1, which modulates the chromatin structure facilitating transcription factors accessibility; and H3K4me3, which is associated with transcriptional start sites of actively transcribed genes. We also observed enrichment for DNase I hypersensitive sites, which highlight regions of open chromatin that are associated to transcriptional activity and transcription factors accessibility.

**Discussion**

In this large-scale GWAS of ease of skin tanning in 176,678 Europeans, we have replicated ten loci previously associated with ease of skin tanning or pigmentation-related phenotypes (BNC2, HERC2/OCA2, IRF4, MC1R, RALY/ASIP, SLC24A4, SLC45A2, TPCN2, TYR and TYRP1), and identified ten novel associations (AHR/AGR3, ATP11A, DCT, EMX2, KIAA0930, PA2G4P4, PDE4B, PPARGC1B, RIPK5 and TRPS1).

The tanning response is determined by an increase in melanin production in melanocytes stimulated by ultraviolet radiation. We identified five novel associations between ease of skin tanning and genes previously suggested to be involved in the melanin synthesis pathway (DCT, EMX2, PDE4B, PPARGC1B and RIPK5)[19–22]. The relative proportion of eumelanin (dark/brown pigment) and phaeomelanin (red/yellow pigment) is regulated by the MC1R gene, which is highly polymorphic in the European population[33]. Different low/intermediate-frequency variants in this gene have been associated with multiple pigmentation-related phenotypes[3,8,9] and both melanoma[11,24,30] and non-melanoma skin cancer[12,13]. Taking advantage of the large UKBB sample size, we tested the association between the nine most studied variants at MC1R[29] and ease of skin tanning. Seven of these variants were highly significantly associated with ease of skin tanning, and they could fully explain our top GWAS association at the MC1R locus.

Non-melanoma skin cancer is the most common type of cancer in the United Kingdom and is well represented in the UKBB data set. Our analyses showed genetic correlation between non-melanoma skin cancer and ease of skin tanning in the UKBB data set. Overall, eight loci associating with ease of skin tanning in our study have been previously shown to influence the susceptibility to either non-melanoma skin cancer (BNC2 and IRF4)[12,13] or both non-melanoma skin cancer and CMM (AHR/AGR3, HERC2/OCA2, MC1R, RALY/ASIP, SLC45A2 and TYR)[11–13]. Additionally, suggestive associations have been previously reported between common variants at ATP11A and TRPS1 and non-melanoma skin cancer[12,13]. Our association for ease of skin tanning at AHR/AGR3 overlapped with the association with CMM identified by Law et al.[11]. Combined analysis of our and Law et al. GWASs results indicated that the same genetic variants at the AGR3/AHR locus were driving the associations at both

phenotypes, thus suggesting that common variants at this locus might increase CMM susceptibility through modulation of ease of skin tanning. Taken together, these results broaden the spectrum of DNA variants involved in ease of skin tanning, and potentially modulating skin cancer risk.

Finally, functional enrichment analysis at the associated DNA variants showed an enrichment for *cis*-eQTLs identified at 5% FDR in sun-exposed skin tissue and fibroblast from the GTEx consortium project[31], and for regulatory elements from the the RoadMap consortium project[32], consistently with the observation that variants identified through GWASs are more likely to act through effects on gene regulation rather than directly altering protein-coding sequences.

## Methods

**Discovery cohort.** The UK Biobank[15] (UKBB) is a prospective cohort study of over 500,000 individuals from across the United Kingdom, aged between 40 and 70. Blood, urine and saliva samples were collected and each participant answered an extensive questionnaire on health and lifestyle. Written informed consent was obtained from each participant, in accordance with the Declaration of Helsinki.

**Genotyping and imputation.** A total of 152,736 samples belonging to the interim data release of UK Biobank genetic data (May 2015) were genotyped using a combination of two arrays: the UK Biobank Axiom array from Affymetrix ($N = 102,754$), that was specifically designed for the purpose of genotyping the UK Biobank participants, and the UK BiLEVE array ($N = 49,982$), that was designed to study the genetics of lung health and disease. The UK10K haplotype reference and the 1000 Genomes Phase 3 reference panels were merged and used as reference panel in the IMPUTE2 software[34]. Kinship coefficients for all pairs were calculated using KING's robust estimator[35] and used to identify and remove related individuals. Imputation, relatedness assessment and quality control were performed by the analysis group at the Wellcome Trust Centre for Human Genetics, University of Oxford. Details are provided at the UK Biobank website (httpλ://biobank.ctsu.ox.ac.uk). A total of 8,351,141 SNPs meeting the following conditions were included in our genome-wide association study: call rate ≥95%, minor allele frequency (MAF) ≥1% and Hardy–Weinberg equilibrium test with $P \geq 1 \times 10^{-9}$.

**Phenotyping.** Ease of skin tanning was collected for 460,922 individuals with self-reported European ethnicity, 140,749 of whom had genotype data. Individuals that reported themselves to be of European ancestry but described their skin colour as 'black' were removed from the data set. As carried out in previous studies of similar phenotypes[3,4] and in order to reduce misclassifications in the self-reported data, individuals reporting that they never tanned only burn, or get mildly or occasionally tanned were included in the group of individuals showing a low tan response, while people who reported getting moderately or very tanned were included in the group of individuals showing a high tan response. Reliability of the reported tanning ability was cross-verified using self-reported information on hair colour, and 2947 individuals reporting themselves to have red hair yet declaring that they get very or moderately tanned were removed from the data set. We further removed 16,067 individuals who were estimated to be genetically related, and 439 individuals showing signs of insufficient data quality, resulting in 121,296 individuals (Supplementary Table 1).

**Association study.** Logistic regression was performed using PLINK[36] (version 1.90 b3.38) assuming an additive genetic model and including sex as covariate, as well as the first five principal components assessed on the genomic data to control for potential population stratification. Associations were considered significant and taken forward for replication if the UKBB discovery *P* value was lower than $5.0 \times 10^{-8}$. Genome-wide Manhattan and Q–Q plots were generated using the *qqman* R package[37] (version 0.1.2). Regional Manhattan plots for the associated loci were generated using *LocusZoom Standalone*[38]. The genotype data from the individuals included in the UKBB data set was used to estimate a more accurate LD structure.

**LD score regression analysis.** The genomic inflation factor ($\lambda_{GC}$) was calculated as the ratio between the observed and expected median $\chi^2$ statistics. We used the LD score regression (LDSC) software[16] (version 1.0.0) to quantify the proportion of such inflation that was due to the presence of polygenic inheritance and to other confounding biases, such as population stratification. Briefly, the LDSC approach evaluates an LD score based on an unbiased estimator of the squared Pearson's correlation, and then regresses the $\chi^2$ statistics against it. The mean contribution of confounding biases in the test statistics is evaluated as the intercept of the regression model minus one. LD scores were evaluated using the 1000 Genomes Project European data[16].

**Identification of independent signals within loci.** We used a stepwise procedure to identify independent signals within the loci identified in the UKBB sample. Specifically, we extended each locus to include a 1 Mb flanking region either side and fitted a new regression model, where the top-associated genome-wide significant SNP was included as a covariate (conditional model). We considered the genome-wide significant ($P < 5 \times 10^{-8}$) top-associated SNP resulting from the conditional model as an independent signal, and included it in the covariate set of a new conditional model. We stopped the stepwise procedure when we could not identify any additional genome-wide significant SNPs. Conditional models were build using PLINK[36] (version 1.90 b3.38).

**Replication cohorts.** The TwinsUK cohort includes more than 13,000 monozygotic and dizygotic twin volunteers from all regions across the United Kingdom[39]. The phenotype was collected via nurse-administered questionnaires using the Fitzpatrick classification[40] and dichotomised into two categories: low (sun-reactive skin type I and II) and high (sun-reactive skin type III and IV) tan response. Phenotypic and genotypic data were available for 3937 female individuals with European ancestry (Supplementary Table 3). Microarray genotyping was conducted using a combination of Illumina arrays (HumanHap300, Human-Hap610, 1M-Duo and 1.2M-Duo 1 M) and imputation was performed using the IMPUTE2 software using haplotype information from the 1000 Genomes Project (Phase 1, integrated variant set across 1092 individuals, v2, March 2012), as previously described[41]. Guy's and St Thomas' Hospital NHS Trust Research Ethics Committee approved the study, and all twins provided informed consent.

The Rotterdam Study (RS) is a prospective study of men and women from the Ommoord municipality of Rotterdam, the Netherlands[42]. Subjects were genotyped using the Infinium II HumanHap550 K and Human 610 Quad Arrays of Illumina. Imputation was performed using the MaCH and minimac software packages and the 1000 Genomes Project (Phase 1, integrated variant set across 1092 individuals, v2, March 2012) as the reference panel, as described elsewhere[43]. Ease of skin tanning was collected on 10,451 individuals via questionnaires, and individuals who reported getting easily burned while in the sun were considered as having a low tan response (Supplementary Table 4). All Rotterdam Study participants have provided written informed consent. The study has been approved by the medical ethics committee according to the Wet Bevolkingsonderzoek ERGO (Population Study Act Rotterdam Study), executed by the Ministry of Health, Welfare and Sports of the Netherlands.

The Australian Study from the Queensland Institute of Medical Research (QIMR) comprises twins and their family members taking part in a long-running study of melanoma risk factors[44]. Two independent samples were used for this analysis. The first included adolescent twins, their siblings and parents, and the second a collection of adult twins. These were genotyped in two phases. Phase 1 samples were genotyped using Illumina Human610-Quadv1_B arrays at deCODE Genetics, Iceland. For imputation, they were merged with a larger set of individuals genotyped on Human610-Quadv1_B, and Human660W-Quad_v1_C arrays. Phase 2 samples were genotyped on HumanOmniExpress-12v1-1_A, HumanOmni25M-8v1-1_B and HumanCoreExome 12v1-0_C arrays from Illumina at the Diamantina Institute, University of Queensland. Self-reported ease of skin tanning was collected on 5509 individuals from the adolescents plus parents, and on 2248 individuals from the adult twins, but only a total of 5149 individuals had genotype data and were included in this analysis (Supplementary Tables 5 and 6). Individuals reporting that they never tanned only burn, or usually burn and sometimes tanned were included in the group of individuals showing a low tan response, while people who reported of usually or always getting tanned were included in the group of individuals showing a high tan response. The study protocol was approved by appropriate institutional review boards, and all participants have provided written informed consent.

The Nurses' Health Study (NHS), the Nurses' Health Study 2 (NHS2) and the Health Professionals Follow-Up Study (HPFS) are three large prospective cohort studies of US men and women of European ancestry (HS). Ease of skin tanning was assessed on 35,845 individuals by asking what kind of tan was developed after repeated sun exposures (e.g., a 2-week vacation outdoors) during childhood or adolescence, and categorised into a binary variable (Supplementary Table 7). Subjects were genotyped on multiple arrays (Affymetrix, Illumina HumanHap, Illumina OmniExpress, HumanCore Exome and OncoArray; Supplementary Table 8) and imputed to approximately 47 million markers using the 1000 Genomes mixed population Project Phase 3 Integrated Release Version 5 (2010–11 data freeze, 2012-03-14 haplotypes) as reference panels. Specifically, SNP genotypes were imputed in two steps. First, genotypes on each chromosome were phased using ShapeIT (v2.r837). Then, phased data were submitted to the Michigan Imputation Server, and imputed using Minimac3. The protocol of the study was approved by the Institutional Review Board of Brigham and Women's Hospital and the Harvard School of Public Health.

**Meta-analysis.** Association studies in the TwinsUK and QIMR cohorts were conducted for individual SNPs using a linear mixed model as implemented in Merlin[45] and GEMMA[46], respectively, in order to take into account the non-independence of twin data. Association studies in the RS and in the HS cohorts were conducted using a logistic regression model adjusting for sex, age and the first five genotype principal components. Specifically, in the HS cohort, associations in

each component GWAS set (Affymetrix, Illumina HumanHap series, Illumina OmniExpress, HumanCore Exome, and OncoArray) were combined in an inverse-variance-weighted meta-analysis using the METAL software[47].

Meta-analyses of the results obtained in the replication cohorts were carried out using a weighted Z-score method based on sample size, $P$ value and direction of effect in each study as implemented in the METAL software[47]. We considered an association replicated when the meta-analysis $P$ value reached a Bonferroni-adjusted significance threshold of $P < 0.05/30 = 1.67 \times 10^{-3}$.

**SNP-by-sex interaction**. We assessed SNP-by-sex interaction in the UKBB data set for all replicated SNPs using PLINK[36] (version 1.90 b3.38). The first five principal components assessed on the genomic data were included as covariates to control for potential stratification issues. We considered an interaction term significant when its $P$ value reached a Bonferroni-adjusted significance threshold of $P < 0.05/20 = 2.5 \times 10^{-3}$. We then attempted replication using four of our replication cohorts (the TwinsUK cohort included only female individuals). The meta-analysis in the replication cohorts was performed using a weighted Z-score method based on sample size, $P$ value and direction of effect in each study as implemented in the METAL software. We considered the interaction replicated when the meta-analysis $P$ value was smaller than the Bonferroni-adjusted significance threshold of $P < 0.05/5 = 0.01$.

**Power calculations**. Power was assessed with Genetic Power Calculator[48] assuming a prevalence of 40% for increased tanning ability—as observed in both the UKBB (42%) and TwinsUK (38%) data sets.

**Identification of known loci**. We interrogated the NHGRI-EBI GWAS catalogue[49] (release 26 June 2017; association $P < 5 \times 10^{-8}$) to identify overlap between SNPs identified in our study (or in tight linkage disequilibrium with them; $r^2 \geq 0.8$) and previously reported associations for ease of skin tanning, pigmentation-related phenotypes and skin cancer.

**Genetic correlation between tanning ability and skin cancer**. The 2017 release of the UK Biobank genetic data included a further 367,186 individuals. We removed 968 individuals flagged because of low genetic data quality, 120,502 individuals who were estimated to be genetically related and 57,157 individuals affected by any cancer, either malignant or in situ, resulting in 907 CMM (ICD-10 code: C43) and 5912 non-melanoma skin cancer (ICD-10 code: C44) cases, and 181,740 controls (Supplementary Table 12). Genotype data were processed as described in 'Geno-typing and imputation', resulting in 5,734,850 SNPs meeting the following conditions: call rate $\geq 95\%$, minor allele frequency (MAF) $\geq 1\%$ and Hardy–Weinberg equilibrium test with $P \geq 1 \times 10^{-9}$. Due to the small number of CMM cases, we only assessed association between ease of skin tanning and non-melanoma skin cancer using a logistic regression approach, as implemented in PLINK (version 2.00), assuming an additive genetic model, and including age, sex, genotyping array and the first five principal components assessed on the genomic data as covariates.

We used the cross-trait LD score regression (LDSC) software[16,50] (version 1.0.0) to estimate the genetic correlation between ease of skin tanning and cancer occurrence. We followed the protocol described in Bulik-Sullivan et al.[50], removing indels, structural variants, strand ambiguous SNPs and those with MAF <1%. LD scores were evaluated using the 1000 Genomes Project European data[16].

**Identification of loci affecting CMM and tanning ability**. We applied a Bayesian bivariate analysis as implemented in GWAS-PW[27] to investigate whether loci here associated with ease of skin tanning and previously involved in melanoma risk exert a shared effect on both ease of skin tanning and CMM, using data from a large meta-analysis of CMM[11] ($N_{case} = 15{,}990$; $N_{control} = 26{,}409$). GWAS-PW estimates the posterior probability that each locus includes a genetic variant that (i) associated only with CMM, (ii) associated only with tanning ability, (iii) associated with both traits or (iv) that the genomic block includes two genetic variants, associating independently with each of the two traits.

**Association study with natural hair colour**. Self-reported hair colour was assessed via questionnaire in 118,777 out of 121,296 individuals used in this study (Supplementary Tables 14 and 15). To assess association between hair colour and the loci replicated in our study of ease of skin tanning, we used (a) a linear regression model to test association with non-red-haired individuals where blonde = 1, light brown = 2, dark brown and black = 3, and (b) a logistic regression model to test association with red versus non-red hair colour. Both linear and logistic regression were performed using PLINK[36] (version 1.90 b3.38) assuming an additive genetic model and including age, sex and the first five principal components assessed on the genomic data as covariates.

**GTEx _cis_-eQTL analysis**. To study whether the replicated SNPs have a regulatory effect on gene expression levels, we used expression quantitative trait loci (eQTLs) data in three skin tissues from the GTEx consortium project[31] (data release v6), namely fibroblasts, sun-exposed skin (lower leg) and non-sun-exposed skin (suprapubic). _cis_-eQTLs were defined by the GTEx consortium as SNPs 1 MB around the transcript start site passing 5% FDR[31]. We extended the set of 34

independent SNPs replicated for ease of skin tanning by including any SNP in high linkage disequilibrium ($r^2 \geq 0.8$) with them using SNAP Proxy Search[51] and data from both the HapMap 3 (release 2) and 1000 Genome (pilot 1) projects. We then evaluated an empirical enrichment $P$ value by comparing the overlap between the set of _cis_-eQTLs in the GTEx project database and the original extended set of SNPs with the overlap obtained using 1000 random sets of SNPs created using a cyclic permutation procedure[52].

**Analysis of epigenetic marks**. By following the same procedure described above for the _cis_-eQTL enrichment analysis, we additionally assessed enrichment and depletion of epigenetic markers by using histone marks and DNA accessibility peak data in epithelial foreskin melanocyte primary cells from the Roadmap consortium project[32] (data release v9). We focused on a core set of histone modifications (namely: H3K4me1, H3K4me3, H3K27me3, H3K36me3, H3K9me3 and H3K27ac) and DNA-seq accessibility data. Since the histone modification data for the studied cell line was available from two donors, we averaged the overlaps among samples.

**URLs**. For UK Biobank, see http://www.ukbiobank.ac.uk/. For LD score regression, see https://github.com/bulik/ldsc. For NHGRI-EBI GWAS catalogue, see https://www.ebi.ac.uk/gwas/. For GWAS-PW, see https://github.com/joepickrell/gwas-pw. For Power calculation, see http://pngu.mgh.harvard.edu/~purcell/gpc/. For SNAP, see http://archive.broadinstitute.org/mpg/snap/. For GTEx portal, see http://www.gtexportal.org/home/. For Roadmap Epigenome Project, see http://www.roadmapepigenomics.org/.

**Data availability**. Supplementary data that support the findings of this study have been deposited in Zenodo with the https://doi.org/10.5281/zenodo.1194289 at the address https://doi.org/10.5281/zenodo.1194289. The association summary statistics for the 10,834 genome-wide significant SNPs are provided as Supplementary Data 1. Genomic coordinates are reported in GRCh37.p13. The association summary statistics for the SNPs associated in the UKBB non-melanoma skin cancer GWAS ($P < 1 \times 10^{-5}$) are provided as Supplementary Data 2. Genomic coordinates are reported in GRCh37.p13. The _cis_-eQTLs identified in the three studied skin tissues, and available within the GTEx project, are provided as Supplementary Data 3.

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

## Acknowledgements

We wish to express our appreciation to all study participants of the UK Biobank, the TwinsUK and the Rotterdam Study cohorts, and to the Australian twin participants and their family members. We thank the participants and staff of the Nurses' Health Study (NHS) and Health Professionals Follow-up Study (HPFS) for their valuable contributions. This work was supported by the Wellcome Trust, grant 081878/Z/06/Z, and the British Skin Foundation, grant 5044i. TwinsUK is funded by the Wellcome Trust, Medical Research Council, European Union (EU), and the National Institute for Health Research (NIHR)-funded BioResource, Clinical Research Facility, and Biomedical Research Centre based at Guy's and St Thomas' NHS Foundation Trust in partnership with King's College London. The Queensland Institute of Medical Research is funded by the NHMRC (103119). The Rotterdam Study is supported by the Erasmus MC and the Erasmus University Rotterdam, the Netherlands Organisation for Scientific Research (NWO), the Netherlands Organisation for Health Research and Development (ZonMw), the Research Institute for Diseases in the Elderly (RIDE), the Netherlands Genomics Initiative (NGI), the Ministry of Education, Culture and Science of the Netherlands, the Ministry of Health Welfare and Sport of the Netherlands, the Municipality of Rotterdam, and the European Commission (DG XII). The Nurses' Health Study (NHS) and Health Professionals Follow-up Study (HPFS) are in part supported by NIH R01 CA49449, P01 CA87969, UM1 CA186107 and UM1 CA167552. F.L. is additionally supported by The China National Thousand Young Talents Award and National Natural Science Foundation of China (NSFC) (91651507). We would like to thank the *Melanoma Meta-analysis Consortium* (a full list of members and affiliations appears in the Supplementary Note) for providing the CMM results to assess shared genetic effect between ease of skin tanning and CMM genes. We wish to acknowledge Julia Sarah El-Sayed Moustafa for useful comments on the manuscript.

## Author contributions

A.V., T.D.S., V.B. and M.F. designed the study. A.V. performed the UKBB GWAS, the meta-analysis, the gene-by-sex interaction, the enrichment analyses and evaluated the genetic correlation between ease of skin tanning and skin cancer. A.V. and M.S. performed the association study in the TwinsUK cohort. F.L., Y.C., C.Z., M.A.H., A.G.U., M.A.I., T.N. and M.K. performed the association study in the RS cohort. G.Z., N.G.M. and D.D. performed the association study in the QIMR cohorts. W.W. and J.H. performed the association study in the NHS and HPFS cohorts. A.V. and P.G.H. performed the association study in natural hair colour. M.M.I., P.A.K. and F.D. provided the CMM results to assess shared genetic effect between ease of skin tanning and CMM genes. D.D. assessed the shared genetic effect between ease of skin tanning and CMM genes. A.V. and M.F. wrote the manuscript. All authors participated in discussion of the final manuscript.

## Additional information

**Competing interests:** The authors declare no competing interests.

