## [Peer Review File · Nature Communications]

Reviewer #1 (Remarks to the Author):

The authors conducted a large scale genome-wide association study (discovery and replication stage) for skin photosensitivity in European subjects and identified both known and novel pigmentation genes. Findings from this study with a large sample size may broaden the spectrum of genetic factors involved in photosensitivity. Overall, the manuscript is written in a good shape and the results presented are informative. I have a few minor comments as follows.

1. The assessment of skin photosensitivity was based on self-reported answers. Did the authors conduct a validation study to validate the accuracy of the self-reported answers?
2. Line 107 – “we defined an expanded set of SNPs including the 7,430 replicated SNPs and those in strong linkage disequilibrium with them.” – what’s the total number of the SNPs included in the expanded set of SNPs?
3. Line 157 - The authors made a conclusion that three previously known melanoma genes act on melanoma risk through modulation of photosensitivity. However, there was no direct evidence provided for this conclusion of “modulation”, i.e., these three melanoma genes was just appeared to be associated with photosensitivity in this study, but no direct statistical evidence was provided for the “conclusion” regarding the effect of three genes on melanoma risk through modulation of photosensitivity in this study.

Reviewer #2 (Remarks to the Author):

This manuscript reported a GWAS on skin photosensitivity. The authors reported some new loci along with some known loci. In general, it is a poorly written manuscript. The phenotype, genetic analysis and the conclusions are very raw and need to shape up substantially. It appears to lack expertise in genetic epidemiology and pigmentation area in the data analysis and manuscript writing. The reviewer has some major comments.

1. The “photosensitivity” is poorly defined. Based on the manuscript, it is the tendency to burn/tan. Burn reflects damaged keratinocytes under apoptosis after acute sun exposure, whereas tanning reflects pigment production after prolonged sun exposure. There is opposite correlation between the two, and very often the questionnaire lists a spectrum from tan to burn. However, the reviewer strongly requests using tanning as much as where possible to define this phenotype. Burn reaction involves other biological mechanism. In addition, the “photosensitivity” in the title and manuscript should be changed to “tendency to tan”.
2. In table 1, there should be an additional column to annotate what’s new loci and what’ known loci.
3. the successful replication is defined as “P from meta is smaller than P from discovery”. It is not an acceptable statistical statement. The replication set has to be nominally significant to claim as successful replication.
4. The abstract and manuscript have mentioned melanoma. But there is no melanoma data in the manuscript to support this speculation. The reviewer requested to add melanoma data and conduct sensitivity and mediation analysis to support their statement.
5. It is not clear how to choose the 9,970 SNPs for replication? Normally, LD prune is applied before moving to the replication phase.
6. The Rotterdam study and Queensland study are very distracting. There is no study description and GWAS QC procedure from both studies. Especially, there is virtually no description on the Rotterdam study. Any GWAS work published on this study? In addition, the phenotype is skin color, which is very different than tanning. Of course, there are correlated. But using their approach to draw the conclusions like shared genetics is very premature. Did they identify additional new loci using combining these two studies? The reviewer does not think the two studies

add any scientific value and should be removed.

7. Phenotype: line 187. It was asked as ordinal variable. Why did they group into binary? A lot of information was lost in this way. Reviewer requested to re-run the analysis to fully use the information available.

8. The QQ plot looks odd. The reviewer suggested to remove the known loci to see the change.

Title: Genome-wide association study in 176,678 Europeans elucidates the genetic architecture of skin sensitivity to sun

Manuscript NCOMMS-16-29474A-Z

Response to the reviewers' comments:

Editor's comments:

You will see that, while the reviewers find your work of interest, they raise substantive concerns that cast doubt on the advance your findings represent over earlier work and the strength of the novel conclusions that can be drawn at this stage. Unfortunately, these reservations are sufficiently important to preclude publication of this study in Nature Communications.

I am sorry that we cannot be more positive on this occasion and thank you for the opportunity to consider your work.

Following the Editorial and Reviewers comments, we have completely revisited and improved our study, increased the replication sample by including an additional >50,000 individuals, and integrated data from the largest melanoma GWAS in collaboration with the Melanoma Meta-analysis Consortium to support the role of skin sensitivity to sun in driving the association between melanoma and the *AHR/AGR3* and *CYP1B1* melanoma genes, thus generating a completely revised submission.

We now include in our study 176,678 European subjects, identifying and replicating 24 genetic loci showing strong and reliable signals of association (with P values up to $\sim 10E^{-500}$), including 13 novel loci for skin sensitivity to sun. Several of the loci identified by our study have been previously associated with increased susceptibility to skin cancer. Please find below out point-to-point answers to the Reviewers comments.

Reviewer #1

1. The assessment of skin photosensitivity was based on self-reported answers. Did the authors conduct a validation study to validate the accuracy of the self-reported answers?

We completely agree with the Reviewer that this is an important consideration. Given the self-reported nature of the phenotype, we cross-validated the reported skin sensitivity to sun answers using data on correlated phenotypes, such as hair and skin colour. Specifically, we removed all individuals that reported themselves to be of Caucasian ancestry but described their skin colour as "black". We also remove individuals describing their hair colour as "red" and reporting that they get very or moderately tanned. We have specified this in the "Phenotyping" section of the Methods: "*Individuals that reported themselves to be of Caucasian ancestry but described their skin colour as "black" were removed from the dataset.*" and "*Reliability of skin sensitivity to sun was cross-verified using self-reported information on hair colour, and 2,947 individuals reporting themselves to have red hair yet declaring that they get very or moderately tanned were removed from the dataset.*"

2. Line 107 – “we defined an expanded set of SNPs including the 7,430 replicated SNPs and those in strong linkage disequilibrium with them.” – what’s the total number of the SNPs included in the expanded set of SNPs?

We agree with the Reviewer on the importance of this information. We updated this figure in order to report the SNPs included in the expanded set of SNPs. Following the suggestion of Reviewer 2, we now include in the replication only a subset of independent SNPs passing genome-wide significance in either the primary or the secondary (conditional) association analyses. The number of independent replicated SNPs is 39 and the number of SNPs included in the expanded set is 717. This is now clarified in the manuscript: *“Finally, we defined an expanded set of SNPs including the 39 replicated SNPs (both primary and secondary associations) and those in strong linkage disequilibrium ($r^2 \geq 0.8$, $N=717$) with them.”*

3. Line 157 - The authors made a conclusion that three previously known melanoma genes act on melanoma risk through modulation of photosensitivity. However, there was no direct evidence provided for this conclusion of “modulation”, i.e., these three melanoma genes was just appeared to be associated with photosensitivity in this study, but no direct statistical evidence was provided for the “conclusion” regarding the effect of three genes on melanoma risk through modulation of photosensitivity in this study.

We have now applied Bayesian bivariate analysis, implemented in GWAS-PW (PMID 27182965), including data from a large meta-analysis on 15,990 cutaneous malignant melanoma (CMM) cases and 26,409 controls from Law *et al* (PMID 26237428). The posterior probabilities for shared genetic effects between skin sensitivity to sun and CMM were 0.985 and 1 for *CYP1B1* and *AGR3/AHR*, respectively, thus suggesting the same genetic variants to be involved in both increased skin sensitivity to sun and increased risk of melanoma. These results have been now added to our manuscript, including a new section *“Identification of genetic loci affecting cutaneous malignant melanoma and skin sensitivity to sun”* in the methods.

While there is an undoubted direction of causality between these two phenotypes, with skin sensitivity to sun being a risk factor for melanoma, more data would be needed to formally test this hypothesis. Therefore, following the Reviewer’s comment we have now modified our sentence as follows: *“Our results suggest that the AHR/AGR3 and CYP1B1 melanoma genes, whose underlying mechanisms are poorly understood, might act on disease risk through modulation of skin sensitivity to sun.”*

Reviewer #2

1. The “photosensitivity” is poorly defined. Based on the manuscript, it is the tendency to burn/tan. Burn reflects damaged keratinocytes under apoptosis after acute sun exposure, whereas tanning reflects pigment production after prolonged sun exposure. There is opposite correlation between the two, and very often the questionnaire lists a spectrum from tan to burn. However, the reviewer strongly requests using tanning as much as where possible to define this phenotype. Burn reaction involves other biological mechanism. In addition, the “photosensitivity” in the title and manuscript should be changed to “tendency to tan”.

We agree with the Reviewer that the term “photosensitivity” should not be used in this context. The questionnaires used to collect both the discovery and replication data used the Fitzpatrick (or a Fitzpatrick-like) scale to measure the skin’s tendency to either tan or burn. To avoid confusion we now follow the same nomenclature used in highly cited GWAS papers previously published by *Nature Genetics* on this phenotype (PMIDs: 17952075, and 18488028), and replaced the word “photosensitivity” with “skin sensitivity to sun” throughout the manuscript.

2. In table 1, there should be an additional column to annotate what’s new loci and what’ known loci.

We agree with the Reviewer that the description of our loci was not very clear. We now highlight the new loci in Figure 1.

3. The successful replication is defined as “P from meta is smaller than P from discovery”. It is not an acceptable statistical statement. The replication set has to be nominally significant to claim as successful replication.

Following the Reviewer’s comment we now declare replication for a locus if the P value in the replication sample is less than the nominal level of significance ($P < 0.05$). This is now reported in the “Meta-analysis” paragraph (Methods) as: “We considered an association replicated if the P value for association in the replication sample was nominally significant ($P < 0.05$) and the locus was tested in all the replication cohorts.”. We have updated the manuscript to reflect the newly-obtained results. In this revisited version of our study we additionally increased the replication sample by including an additional >50,000 individuals.

4. The abstract and manuscript have mentioned melanoma. But there is no melanoma data in the manuscript to support this speculation. The reviewer requested to add melanoma data and conduct sensitivity and mediation analysis to support their statement.

We agree. Despite the associations being at the same loci and them having been detected for two correlated traits, it may still be possible that two distinct genetic variants associate independently with each of the two traits. Consequently, we have now applied the Bayesian bivariate analysis implemented in GWAS-PW including data from a large meta-analysis on 15,990 cutaneous malignant melanoma cases and 26,409 controls from Law *et al* (PMID 26237428). The posterior probabilities for shared genetic effects between skin sensitivity to sun and CMM were 0.985 and 1 for *CYP1B1* and *AGR3/AHR*, respectively, thus suggesting the same genetic variants to be involved in both increased skin sensitivity to sun and increased risk of melanoma. We have now added these results to our manuscript, and a new section “Identification of genetic loci affecting cutaneous malignant melanoma and skin sensitivity to sun” in the methods.

5. It is not clear how to choose the 9,970 SNPs for replication? Normally, LD prune is applied before moving to the replication phase.

We now took forward for replication only 52 independent SNPs discovered through primary and secondary (conditional) analyses, as described in the “Identification of independent signals within loci” paragraph (Methods), and using the procedure implemented in PLINK. We also updated the main text reporting: “After selecting 52 independent signals (Methods), we sought replication in five additional cohorts of European ancestry ($N=55,382$; Methods; Supplementary Tables 3-8), for which data

on skin sensitivity to sun was available. Meta-analysis of the results in the five replication cohorts confirmed the association at 39 of the 52 selected SNPs, and for 24 of the 30 genetic loci across 17 chromosomes, encompassing eleven genes previously associated with skin sensitivity to sun and pigmentation-related traits, and thirteen novel associations (Table 1, Supplementary Table 9, Supplementary Figures 3-26).”

6. The Rotterdam study and Queensland study are very distracting. There is no study description and GWAS QC procedure from both studies. Especially, there is virtually no description on the Rotterdam study. Any GWAS work published on this study? In addition, the phenotype is skin color, which is very different than tanning. Of course, there are correlated. But using their approach to draw the conclusions like shared genetics is very premature. Did they identify additional new loci using combining these two studies? The reviewer does not think the two studies add any scientific value and should be removed.

We agree with the Reviewer that the use of different phenotypes in two of our replication cohorts was indeed distracting. We are now using the Fitzpatrick (or a Fitzpatrick-like) scale to measure the skin’s tendency to either tan or burn in both the discovery and replication cohorts, thus guaranteeing the homogeneity of the phenotypes.

We also now include additional information on the Rotterdam Study, with appropriate references. Specifically, The “Replication cohorts” paragraph (Methods) now reports: “The Rotterdam Study (RS) is a population-based prospective study³⁰. Microarray genotyping was conducted using the Infinium II HumanHap550 K and Human 610 Quad Arrays of Illumina and genotypes were imputed using the 1000-Genomes Project as the reference panel (Phase 1, integrated variant set across 1092 individuals, v2, March 2012) using the MaCH and minimac software packages³¹.”

7. Phenotype: line 187. It was asked as ordinal variable. Why did they group into binary? A lot of information was lost in this way. Reviewer requested to re-run the analysis to fully use the information available.

We agree with the Reviewer’s comments, as we also debated at the beginning of this study whether this self-reported phenotype should have been analysed as an ordinal variable or after dichotomisation. Indeed, data collected through questionnaires are often prone to misclassification since they are influenced by a subjective versus objective assessment. Thus, we decided to build on the experience of previous highly-cited GWAS studies on skin sensitivity to sun data collected through questionnaires and previously published in *Nature Genetics* (PMIDs: 17952075, and 18488028). As reported by the authors of both manuscripts: “The benefits of the self-reported measurements are that they are cheap and easy to collect, but their subjective nature is likely to introduce misclassifications, leading to a loss of power in the discovery phase and a decrease in prediction accuracy”. By following the same approach we also dichotomised the trait in order to increase the signal-to-noise ratio. Following the Reviewer’s comment we additionally re-run the analyses in the UK Biobank dataset using the ordinal variable as response. Results are reassuringly similar, with a correlation in the $-\log_{10}(\text{pvalues})$ of the genome-wide significant results of 0.987.

We now add in the Methods (“Phenotyping” section): “As carried out in previous studies of skin sensitivity to sun^{6,23}, in order to reduce potential misclassifications in the self-reported phenotype, individuals reporting that they never tanned only burn, or get mildly or occasionally tanned were included in the group of sun-sensitive

individuals, while people who reported getting moderately or very tanned were included in the group of non-sun-sensitive individuals."

8. The QQ plot looks odd. The reviewer suggested to remove the known loci to see the change.

We agree with the reviewer that the Quantile-Quantile plot reported in Supplementary Figure 1 may be misleading. In fact, P values were limited to 5×10^{-324} due to the minimum precision allowed by R and the *qqman* package. Now we report an additional Quantile-Quantile plot (Supplementary Figure 2) where we plot the observed *versus* expected P values after removing loci previously associated with skin sensitivity to sun. Specifically, we removed SNPs at the loci harboring the genes *HERC2/OCA2*, *IRF4*, *MC1R*, *RALY/ASIP*, *SLC45A2*, and *TYR*.

Reviewer #1 (Remarks to the Author):

The authors have well addressed the reviewer's comments. The reviewer has no additional comments.

Reviewer #3 (Remarks to the Author):

The manuscript, "Genome-wide association study in 176,678 Europeans elucidates the genetic architecture of skin sensitivity to the sun" by Visconti and colleagues presents a GWAS for tanning. The authors present a set of new loci in addition to identifying loci already known to affect pigmentary traits and skin response phenotypes.

A big question with this study is the definition of the phenotype. Unfortunately, for both the research community interested in pigmentation and skin response to UVR, the Fitzpatrick scale is widely used. It's unfortunate because it confounds two types of responses to the sun (burning and tanning) and skin pigmentation all in one measure. In fact, it's known that they two skin responses are positively correlated when skin color is considered. Thus, by simply changing "photosensitivity" to "skin sensitivity to the sun" (which was done for most but not all instances), I don't think the authors have addressed the issue brought up in the first review. Probably what these authors have measured might best be described as "ease of tanning" as the authors do in the methods. Many will think that "sun sensitivity" is the erythema response, which is not what these authors are investigating.

The replication testing should also include multiple testing adjustments. The appropriate p-value threshold, given any of the loci tested would be considered a finding, should be $0.05/N$, where N is the number of loci tested for replication. In this case, $0.05/24=0.0021$. I would caution against putting any emphasis on those loci that show replication p-values above this threshold. Perhaps those loci that do not replicate could be included in the SOM at most. I think the authors only lose three of the 24 loci discovered by doing so.

I suspect that the single most significant (statistically and biologically) result in this paper has been totally overlooked: Namely the difference between men and women in "sun sensitivity". Although this sex difference has been previously noted, it is not generally well appreciated. Indeed, I calculated the chi-square for the UKBB study from SOM Table 1 to be 2,461.1, which is highly significant for a 2x2 table. The Rotterdam also shows the same pattern of sex difference (chi-square = 77.6) excepting the QIMR sample, which is much younger than the others.

European-derived men have a stronger erythema dose response to UVR and have lighter skin than European-derived women. Given this, it's very reasonable to suppose that some of the loci affecting these traits have sex-dependent patterns of effect. These authors report using sex as a covariate in their GWAS but say nothing more about the sex differences they observed. Given the very large

sample size and the highly significant effects noted for some of the genes (viz., MC1R, RALY/ASIP, INF4, and SLC45A2) have an excellent opportunity here to look for a sex by gene interaction, which could make the report much more substantial.

It would be informative to include the scatter plots of the five PCs the authors used to adjust for variation in ancestry.

I see the authors mention having added an additional QQ plot, but it doesn't appear in the SOM materials file I was able to download. It would be nice the revised QQ plot.

Minor points:

Line 50-51, "large differences within and between populations" – What's a difference within a population? Perhaps the authors mean "variability"?

Line 66, "...highlighted the presence of..." is vague.

Line 68, "...we sought replication in..." is awkward.

Line 162, "Results presents are from..." needs fixing.

Line 203, "Caucasian ancestry" What do the authors mean by this word?

Reviewer #4 (Remarks to the Author):

Review of "Genome-wide association study in 176,678 Europeans elucidates the genetic architecture of skin sensitivity to sun"

Major comments :

1/Gwas study have been performed in the literature on mutiple pigmentation traits. Hair colours, Eye Colours, Skin sentivity to sun and freckling abundance. There is clear genetic overlap between these traits as demonstrated in figure 1 by Sulem et al in NAT genetics 2007.

In the abstract and through the text of the current manuscript , in term of novel loci a distinction should be made between locus never reported for a pigmentation trait in gwas and the one already having such report in gwas (SLC24A4, TYRP1, KITLG, TPCN2).

Then it will create three classes based on novelty.

a/Not known for skin sensitivity

b/Known for another pigmentation trait

c/known for skin sensitivity

As previously suggested by a reviewer in a previous round, the reader will benefit immensely in table 1 of an indication of novelty as described above for each signal/SNP.

2/ Linked to the first comment it is of interest to think of the pigmentation traits together, here it is somewhat in isolation. In extenso, it would have been nice to see the effect of these variants on other pigmentation traits.

We understand that it could be out of the scope of this paper in principle but given the shared genetic predisposition and given

Minor :

1-Title : Page 1 line 3

The word "elucidates" is too strong. A lot was known before their work.

2- Abstract: Page 1 line 40

"genetic loci" are not themselves associated they contain sequence variants that associate.

"13 novel associations are claimed", it would be interesting to know how many unique novel loci are added.

3-Abstract: Page 1 line 41 "AHR/AGR3 and CYP1B1" are not melanoma genes, they are loci with sequence variants associated to melanoma.

4-Page 1 line 52: genes are usually not identified by GWAS, but rather sequence variants associating to disease that can lead to specific gene by coding/splicing effect, expression QTL, or proximity.

5-Page 2 line 69 to 74. Sentence is very long as well as the overall chapter.

Might be better to represent a diagram or schematic with counts

6-Page 1 line 55. It is unclear how the phenotype is distributed and treated. It appears in some supplement that in the UKB group, the information is dichotomized (sun sensitive/ non sun sensitive). This should appear in the text and the fraction of the two groups should be added.

7- Page 2 line 79-80 when sequence variants at loci previously reported with eye or hair colour, it would be informative to have a feeling that the sun sensitivity association is much weaker than the first reported traits.

8-Page 2 , when discussing the novel locus: it should be mentioned that they all are much less significant. They are all common except the AHR/AGR3 locus and more finding seems to be related to bigger sample size rather than a more exhaustive coverage of genetic diversity.

9- At MC1R locus a well published locus, one would have expected to have a better representation of the multiple reported signals. A 30% variant is in the report, but it is well established that it is one of the rare cases of syntenic association (a more frequent variant, 30%, than the causative ones, covering mainly two of them of less frequency each circa 10%)

Reviewers' comments:

Reviewer #1 (Remarks to the Author):

The authors have well addressed the reviewer's comments. The reviewer has no additional comments.

Reviewer #3 (Remarks to the Author):

The manuscript, "Genome-wide association study in 176,678 Europeans elucidates the genetic architecture of skin sensitivity to the sun" by Visconti and colleagues presents a GWAS for tanning. The authors present a set of new loci in addition to identifying loci already known to affect pigimentary traits and skin response phenotypes.

A big question with this study is the definition of the phenotype. Unfortunately, for both the research community interested in pigmentation and skin response to UVR, the Fitzpatrick scale is widely used. It's unfortunate because it confounds two types of responses to the sun (burning and tanning) and skin pigmentation all in one measure. In fact, it's known that they two skin responses are positively correlated when skin color is considered. Thus, by simply changing "photosensitivity" to "skin sensitivity to the sun" (which was done for most but not all instances), I don't think the authors have addressed the issue brought up in the first review. Probably what these authors have measured might best be described as "ease of tanning" as the authors do in the methods. Many will think that "sun sensitivity" is the erythema response, which is not what these authors are investigating.

Answer: We understand the concern of this Reviewer and now define the studied phenotype as 'ease of skin tanning' or 'tanning ability' throughout the manuscript.

The replication testing should also include multiple testing adjustments. The appropriate p-value threshold, given any of the loci tested would be considered a finding, should be $0.05/N$, where N is the number of loci tested for replication. In this case, $0.05/24=0.0021$. I would caution against putting any emphasis on those loci that show replication p-values above this threshold. Perhaps those loci that do not replicate could be included in the SOM at most. I think the authors only lose three of the 24 loci discovered by doing so.

Answer: We agree with the Reviewer that a Bonferroni-corrected threshold should be used. In this revised version of the manuscript, we focus only on those SNPs passing a Bonferroni-adjusted threshold of $0.05/30=0.00167$, which takes into account the fact that we attempted to replicate 30 loci identified in the UKBB cohort. Using this threshold, the number of replicated loci decreases to 20, with four loci no longer replicated and encompassing the genes *CYP1B1*, *FOXD3*, *KITLG*, and *PDE4D*.

We have now modified the Methods section: "We considered an association replicated when the meta-analysis P value reached a Bonferroni-adjusted significance threshold of $P<0.05/30=1.67\times 10^{-3}$." We have also updated the main text accordingly. SNPs that reached nominal significance in the replication step are now reported in Supplementary Table 9.

I suspect that the single most significant (statistically and biologically) result in this paper has been totally overlooked: Namely the difference between men and women

in “sun sensitivity”. Although this sex difference has been previously noted, it is not generally well appreciated. Indeed, I calculated the chi-square for the UKBB study from SOM Table 1 to be 2,461.1, which is highly significant for a 2x2 table. The Rotterdam also shows the same pattern of sex difference (chi-square = 77.6) excepting the QIMR sample, which is much younger than the others. European-derived men have a stronger erythema dose response to UVR and have lighter skin than European-derived women. Given this, it’s very reasonable to suppose that some of the loci affecting these traits have sex-dependent patterns of effect. These authors report using sex as a covariate in their GWAS but say nothing more about the sex differences they observed. Given the very large sample size and the highly significant effects noted for some of the genes (viz., MC1R, RALY/ASIP, INF4, and SLC45A2) have an excellent opportunity here to look for a sex by gene interaction, which could make the report much more substantial.

Answer: This is a very interesting observation, and we consequently ran a sex-by-SNP interaction model. In the UKBB dataset five of the 20 replicated SNPs showed significant interaction ($P < 0.05/20 = 2.5 \times 10^{-3}$). However, none of them survived the replication step.

We have now added the following paragraph to the Method Section:

“SNP-by-sex interaction. We assessed SNP-by-sex interaction in the UKBB dataset for all replicated SNPs using PLINK³¹ (version 1.90 b3.38). The first five principal components assessed on the genomic data were included as covariates to control for potential stratification issues. We considered an interaction term significant when its P value reached a Bonferroni-adjusted significance threshold of $P < 0.05/20 = 2.5 \times 10^{-3}$. We then attempted replication using four of our replication cohorts (the TwinsUK cohort included only female individuals). The meta-analysis in the replication cohorts was performed using a weighted Z-score method based on sample size, P value and direction of effect in each study as implemented in the METAL software. We considered the interaction replicated when the meta-analysis P value was smaller than the Bonferroni-adjusted significance threshold of $P < 0.05/5 = 0.01$.”

and discussed it in the main text as:

*“Given the large sample size and the highly significant effects identified in this study, we further investigated the sex-by-gene interaction for the 20 known and novel loci here associated with tanning ability. Indeed, it has been repeatedly observed that adult females are fairer-skinned than males¹⁷. We identified significant interactions ($P < 2.5 \times 10^{-3}$) at five different loci in the UKBB dataset, which, however, were not confirmed in the additional cohorts (**Supplementary Table 10**).”*

Results are reported as Supplementary Table 10.

It would be informative to include the scatter plots of the five PCs the authors used to adjust for variation in ancestry.

Answer: We now add the PC plots as Supplementary Figure 1.

I see the authors mention having added an additional QQ plot, but it doesn’t appear in the SOM materials file I was able to download. It would be nice the revised QQ plot.

Answer: We apologize for the confusion. The additional QQ plot is now available as Supplementary Figure 3.

Minor points:

Line50-51, “large differences within and between populations” – What’s a difference within a population? Perhaps the authors mean “variability”?

Answer: We agree that *variability* is a more precise term, and we now write: “*The tanning response after exposure to sunlight shows large variability within and between populations, and one of its main determinants is melanin pigmentation.*”

Line 66, “...highlighted the presence of...” is vague.

Answer: We agree with this Reviewer and now write: “*Therefore, to rule out the possibility that some associations could be driven by confounding biases, we used the LD-score regression approach⁵, which confirmed the presence of an underlying polygenic architecture (intercept=1.04±0.01).*”

Line 68, “...we sought replication in...” is awkward.

Answer: We modified the sentence, which is now: “*We attempted to replicate these 30 loci in five additional cohorts of European ancestry for which data on tanning ability was available.*”

Line 162, “Results presents are from...” needs fixing.

Answer: We changed the Figure’s caption as: “*The P values were obtained by logistic regression analysis assuming an additive genetic model with sex and the first five principal components of the genotype data as covariates.*”

Line 203, “Caucasian ancestry” What do the authors mean by this word?

Answer: We agree with the Reviewer that the word “*Caucasian*” may be misleading and we now use “*European ancestry*”: “*Individuals that reported themselves to be of European ancestry but described their skin colour as “black” were removed from the dataset.*”

Reviewer #4 (Remarks to the Author):

Review of "Genome-wide association study in 176,678 Europeans elucidates the genetic architecture of skin sensitivity to sun"

Major comments :

1/Gwas study have been performed in the literature on mutliple pigmentation traits. Hair colours, Eye Colours, Skin sentivity to sun and freckling abundance. There is clear genetic overlap between these traits as demonstrated in figure 1 by Sulem et al in NAT genetics 2007.

In the abstract and through the text of the current manuscript , in term of novel loci a distinction should be made between locus never reported for a pigmentation trait in gwas and the one already having such report in gwas (SLC24A4, TYRP1, KITLG, TPCN2).

Answer: We agree that the distinction between loci is not clear in both the abstract and the main text of the manuscript. In the abstract we now report: “*We identify significant association with tanning ability at 20 loci. We confirm previously identified associations at six of these loci, and report 14 novel loci, of which ten have never been associated with pigmentation-related phenotypes.*”

We have also changed the main text as: “The replicated loci encompass six genes previously associated with tanning ability, four genes previously associated with pigmentation-related traits, and ten novel associations (**Table 1, Supplementary Tables 2 and 9, Supplementary Figures 4-23**).” We also discuss the results in the same manner, that is, first reporting the known associations for tanning ability, then the loci known to be associated with pigmentations related phenotypes, and finally the novel associations.

Then it will create three classes based on novelty.

a/Not known for skin sensitivity

b/Known for another pigmentation trait

c/known for skin sensitivity

As previously suggested by a reviewer in a previous round, the reader will benefit immensely in table 1 of an indication of novelty as described above for each signals/SNP.

Answer: We now report the class of novelty of the association in Table 1 and in Figure 1.

2/ Linked to the first comment it is of interest to think of the pigmentation traits together, here it is somewhat in isolation. In extenso, it would have been nice to see the effect of these variants on other pigmentation traits.

We understand that it could be out of the scope of this paper in principle but given the shared genetic predisposition and given

Answer: We agree with this Reviewer that a comparison of the genetic effects of our replicated loci on other pigmentation traits can be interesting, and that it may also help to understand whether they are specific for tanning ability or more broadly associated with pigmentation traits. Of the traits studied by Sulem et al. (2008), namely freckles, eye and hair colour, only the natural hair colour was collected by the UK Biobank. A GWAS for the hair colour phenotype in the UK Biobank has recently been carried out and independently replicated in a very large meta-analysis involving 23andMe (Hysi et al, manuscript currently under advanced stages of revision). We have been granted access to the replicated results from Hysi et al, which we use to validate our observations on novel loci associating with hair colour (and add a reference to Hysi et al to our manuscript).

We now add the following paragraphs in the Method section:

“Association study with natural hair colour. Self-reported hair colour was assessed via questionnaire in 118,777 out of 121,296 individuals used in this study (**Supplementary Table 14 and 15**). To assess association between hair colour and the loci replicated in our study of ease of skin tanning, we used a) a linear regression model to test association with non-red haired individuals where blonde=1, light brown=2, dark brown and black=3, and b) a logistic regression model to test association with red versus non-red hair colour. Both logistic and linear regression were performed using PLINK³¹ (version 1.90 b3.38) assuming an additive genetic model and including age, sex and the first five principal components assessed on the genomic data as covariates..”

We add the following paragraph to the main text:

“We further investigated the effects of known and novel loci here associated with tanning ability on hair colour, which was also characterised in the UKBB. Particularly, we separately assessed the association with non-red hair colour, and with red versus non-red hair colour (Methods, Supplementary Table 14 and 15**). Most of the genes that have been previously associated with non-red hair colour^{9,23,25} (i.e., IRF4, HERC2/OCA2, SLC24A4, SLC45A2, TPCN2, and TYRP1) showed a stronger**

association with non-red hair colour than with ease of skin tanning, with the exception of MC1R, RALY/ASIP and TYR (**Supplementary Table 16**). We additionally observed three novel genome-wide significant associations between non-red hair colour and BNC2 (previously associated with facial pigmentation⁹ and freckles²³), DCT, and RIPK5, which were confirmed and replicated in a large meta-analysis by Hysi et al. (manuscript under revision). A marginal association was observed at KIAA0930. Red hair showed significant association with IRF4, HERC2/OCA2, MC1R, and RALY/ASIP, while a marginal association was observed at SLC45A2 (**Supplementary Table 17**). Only MC1R was more strongly associated with red hair colour than with ease of skin tanning. Overall, seven loci were exclusively associated with tanning ability (AHR/AGR3, ATP11A, EMX2, PA2G4P4, PDE4B, PPARGC1B, and TRPS1).”

We further compared the effect of the nine most-studied MC1R variants on both ease of skin tanning and hair colour (please see your minor point 9).

Minor :

1-Title : Page 1 line 3

The word "elucidates" is too strong. A lot was *known before their work*.

Answer: Following this Reviewer's comment, we have now modified the title as follows: *Genome-wide association study in 176,678 Europeans reveals new genetic loci for tanning response to sun exposure*

2- Abstract: Page 1 line 40

"genetic loci" are not themselves associated they contain sequence variants that associate.

Answer: We now report: *"We identify significant association with tanning ability at 20 loci."*

"13 novel associations are claimed" , it would be interesting to know how many unique novel loci are added.

Answer: We now updated the abstract to clarify this point and write: *"We identify significant association with tanning ability at 20 loci. We confirm previously identified associations at six of these loci, and report 14 novel loci, of which ten have never been associated with pigmentation-related phenotypes."*

3-Abstract: Page 1 line 41 *"AHR/AGR3 and CYP1B1" are not melanoma genes, they are loci with sequence variants associated to melanoma.*

Answer: We have now edited the sentence in the abstract and use a more precise definition: *"Our results also suggest that variants at the AHR/AGR3 locus, previously associated with cutaneous malignant melanoma and whose underlying mechanism is poorly understood, might act on disease risk through modulation of tanning ability."*

4-Page 1 line 52: *genes are usually not identified by gwas, but rather sequence variants associating to disease that can lead to specific gene by coding/splicing effect. expression qtl, or proximity.*

Answer: We agree with the reviewer that the word "genes" may be misleading and we now use "DNA variants": *"Several DNA variants involved in tanning ability and pigmentation have been identified by genome-wide association studies"*

5-Page 2 line 69 to 74 .Sentence is very long as well as the overall chapter.
Might be better to represent a diagram or schematic with counts

Answer: We agree that the previous sentence was very long, and we now split it in three shorter sentences: “We attempted to replicate these 30 loci in five additional cohorts of European ancestry for which data on tanning ability was available (N=55,382; **Methods, Supplementary Tables 3-8**). Meta-analysis of the results in these five replication cohorts confirmed, at a Bonferroni-corrected threshold of $0.05/30=1.67\times 10^{-3}$, the association at 20 of the 30 top-associated SNPs at each locus. The replicated loci encompass six genes previously associated with tanning ability, four genes previously associated with pigmentation-related traits, and ten novel associations (**Table 1, Supplementary Tables 2 and 9, Supplementary Figures 4-23**).”.

6-Page 1 line 55. It is unclear how the phenotype is distributed and treated. It appears in some supplement that in the UKB group, the information is dichotomized (sun sensitive/ non sun sensitive). This should appear in the text and the fraction of the two group should be added.

Answer: We agree with this Reviewer that this is an important piece of information and should appear in the main text and not only in the Methods and Supplementary Material. We now write: “Ease of skin tanning and genotype data were available for 121,296 individuals of European ancestry from the UKBB which were divided in two groups according to their skin’s ability to tan, with 38.6% of the individuals reporting that they never tan and only burn, or get mildly or occasionally tanned (**Methods; Supplementary Table 1**).”.

7- Page 2 line 79-80 when sequence variants at loci previously reported with with eye or hair colour , it would be informative to have a feeling that the sun sensisitivity association is much weaker that the first reported traits.

Answer: Please see our answer to your major point 2.

8-Page 2 , when discussing the novel locus: it should be mentionned that they all are much less significant. They are all common except the AHR/AGR3 locus and more finding seems to be related to bigger sample size rather that a more exhaustive coverage of genetic diversity.

Answer: We agree with this Reviewer that our power calculation could be misleading. Indeed, the newly discovered associations are with common SNPs and in general show smaller effects than the already-known loci. We have now changed the sentence into: “The high statistical power provided by the large UKBB sample size (>80% for a common variant with minor allele frequency of 0.5 and odds ratio > 1.06 at α -level 5×10^{-8}) allowed for the identification of low-penetrance common DNA variants of small effect size at ten novel genes.”.

9- At MC1R locus a well published locus, one would have expected to have a better representation of the mutliple rported signals. A 30% variant is in the report, but is well established that it is one of the rare case of syntenic association (a more frequent variant ,30%, that the causative ones, cover mainly two of them of less freq each circa 10%)

Answer: We agree with the Reviewer, and along with the leading variant reported in Table 1, we now report in Supplementary Table 15 and 16 the summary statistics for

the nine most studied *MC1R* variants (e.g. PMID:18366057) available in our panel in both ease of skin tanning and hair colour.

We now write in the main text: “*The MC1R gene regulates the relative proportion of eumelanin (dark/brown pigment) and pheomelanin (red/yellow pigment), and is highly polymorphic in the European population*²⁴. Different low/intermediate-frequency variants in this gene have been associated with multiple pigmentation-related phenotypes^{9,23,25} and both melanoma^{20,26} and non-melanoma skin cancer^{2,3}. To better characterise the *MC1R* gene, we took advantage of the large UKBB sample size and tested the association between the nine most studied *MC1R* variants²⁷ present in our panel (D84E, D294H, I155T, R142H, R151C, R160W, R163Q, V60L, and V92M) and both ease of skin tanning and natural hair colour. Among these, the GWAS Catalog reports association between R151C and ease of skin tanning, freckles, hair and skin colour^{8,25}, and both melanoma²⁶ and non-melanoma skin cancers^{2,3}. A haplotype including multiple *MC1R* variants was also associated with skin pigmentation²⁸. In the UKBB dataset, all variants apart from I155T, were strongly associated with red hair colour, and all variants but R163Q were associated with non-red hair (**Supplementary Table 18**). Moreover, all the studied nine variants but two (R163Q and V92M) were highly significantly associated with ease of skin tanning (**Supplementary Table 19**). Interestingly, these variants at the *MC1R* gene completely explained our top-associated signal at rs369230, which changed from $P=8.28 \times 10^{-132}$ to 0.37.”.

Reviewer #4 (Remarks to the Author):

The author have now adressed the reviewer comments.

The paper is a much more pleasant read.

Reviewer #5 (Remarks to the Author):

I am satisfied with the author responses to reviewer #3 comments.

The one comment I would make is that, as with many self-reported UK Biobank phenotypes, it can be quite unclear what aspect of the trait the variable is actually assessing. For example, CADM2 appears significantly associated in the manhattan plot and is a well known locus for risk-taking behavior. I think it would be beneficial to directly and comprehensively (i.e genome-wide) show the relevance of this analysed phenotype to skin cancer (rather than just a few example loci). This could be done via a genetic correlation (LDSC) or a cross-trait meta-analysis (MTAG) to demonstrate the benefit of assessing this trait. The full release of UKBB should have more than enough cancer cases to enable these analyses.

Response to Reviewers' comments:

Reviewers' comments:

Reviewer #5 (Remarks to the Author):

I am satisfied with the author responses to reviewer #3 comments.

The one comment I would make is that, as with many self-reported UK Biobank phenotypes, it can be quite unclear what aspect of the trait the variable is actually assessing. For example, CADM2 appears significantly associated in the manhattan plot and is a well known locus for risk-taking behavior. I think it would be beneficial to directly and comprehensively (i.e genome-wide) show the relevance of this analysed phenotype to skin cancer (rather than just a few example loci). This could be done via a genetic correlation (LDSC) or a cross-trait meta-analysis (MTAG) to demonstrate the benefit of assessing this trait. The full release of UKBB should have more than enough cancer cases to enable these analyses.

Answer: We agree with the Reviewer that multiple source of noise may affect self-reported data. However, we are confident that the replication step allowed us both to discard artefacts due to uncontrolled biases and to confirm credible associations. Indeed, CADM2, despite being detected as significantly associated in the UKBB, failed replication, not even reaching nominal significance in meta-analysis.

A better support for the relevance of ease of skin tanning to skin cancer would be to show that these traits are actually genetically correlated, as suggested by the Reviewer.

The incidence of both melanoma and non-melanoma skin cancer is related to age, with highest incidence rate after the age of 85. The mean (median) age distribution in UKBB is approximately 57 (58), ranging between 38 and 77, therefore a limited number of CMM and non-melanoma skin cancer cases are expected in the UKBB sample. Using the 2017 release of the UK Biobank genetic data we identified 907 melanoma cases, therefore our study is underpowered to assess the genetic correlation between CMM and tanning response. The number of non-melanoma skin cancer was 5,912, in line with the expected higher incidence of this disease in the general population (up to 8 times higher than melanoma's incidence, by considering the UKBB age range and based on the statistics published by Cancer Research UK).

We now add the following paragraphs in the Method section:

“Genetic correlation between ease of skin tanning and skin cancer. *The 2017 release of the UK Biobank genetic data included a further 367,186 individual. We removed 968 individuals flagged because of low genetic data quality, 120,502 individuals who were estimated to be genetically related, and 57,157 individuals affected by any cancer, either malignant or in situ, resulting in 907 CMM (ICD-10 code: C43) and 5,912 non-melanoma skin cancer (ICD-10 code: C44) cases, and 181,740 controls (Supplementary Table 10). Genotype data were processed as described in “Genotyping and imputation”, resulting in 5,734,850 SNPs meeting the following conditions: call rate $\geq 95\%$, minor allele frequency (MAF) $\geq 1\%$ and Hardy-Weinberg equilibrium test with $P \geq 1 \times 10^{-9}$. Due to the small number of CMM cases, we only assessed association between ease of skin tanning and non-melanoma skin cancer using a logistic regression approach, as implemented in PLINK (version 2.00), assuming an additive*

genetic model, and including age, sex, genotyping array, and the first five principal components assessed on the genomic data as covariates.

We used the cross-trait LD-score regression (LDSC) software^{5,45} (version 1.0.0) to estimate the genetic correlation between ease of skin tanning and cancer occurrence. We followed the protocol described in Bulik-Sullivan et al.⁴⁵, removing indels, structural variants, strand ambiguous SNPs, and those with MAF < 1%. LD scores were evaluated using the 1000 Genomes Project European data⁵.”

We add the following paragraph to the Results Section:

*“While the number of melanoma cases in the UKBB was not sufficient to generate a reliable estimate of the genetic correlation between ease of skin tanning and CMM, we could estimate the genetic correlation of ease of skin tanning with non-melanoma skin cancer, which amounted to $\rho=0.33$ ($SD=0.16$, $P=0.04$; **Methods, Supplementary Table 10 and 11, Supplementary Figures 24 and 25, Supplementary Data 2).**”*

Reviewer #5 (Remarks to the Author):

I am satisfied by the author responses to my comment.